# Toward robots that learn to summarize their actions in natural language: a set of tasks

**Chad DeChant**
Columbia University
chad.dechant@columbia.edu

**Daniel Bauer**
Columbia University
bauer@cs.columbia.edu

**Abstract:** Robots should be able to report in natural language what they have done. They should provide concise summaries, respond to questions about them, and be able to learn from the natural language responses they receive to their summaries. We propose that developing the capabilities for robots to summarize their actions is a new and necessary challenge which should be taken up by the robotic learning community. We propose an initial framework for *robot action summarization*, presented as a set of tasks which can serve as a target for research and a measure of progress.

**Keywords:** Explainable robotics, Long horizon tasks, Summarization

## 1 Introduction

Humans often tell each other what we've done. We talk about the distant past as well as things that happened just a few minutes ago. We tell a co-worker what part of our shared task we finished, what remains to be done, and why. Children talk about their day at school, gradually learning how to construct a coherent narrative of events and providing parents an opportunity to offer positive or negative feedback as well as advice. Robots should be learning to do the same.

Robots are increasingly able to perform complex, long horizon tasks [1, 2]. The fact that long horizon tasks take place over an extended period of time and potentially in many locations makes it difficult both to give robots instructions and to monitor what actions they subsequently perform. Much work has been done to enable robots to learn how to follow natural language instructions, including how to translate high-level commands into low-level actions and planning [3]. We suggest that such work is only half of a necessary communication cycle and that it is time for the robotic learning community to tackle what might be considered the inverse task of enabling robots to summarize what they have done in succinct natural language. This ability would be especially important for long horizon tasks during which constant user attention and supervision is unrealistic and undesirable. For example, a home assistant robot may spend hours cleaning; its user would not want to monitor its behavior nor would he want a complete report of everything the robot did. But the user would likely want a quick summary of which rooms were cleaned successfully, what remains to be cleaned, and if any problems were encountered during the robot's operation.

We propose a set of tasks to provide goals to work towards and to illustrate the benefits of robots' being able to summarize their actions in natural language. After starting with the pure summarization task, we suggest that such summaries could be enriched with explanations of actions, could prompt questions from a user which the robot should be able to answer, and could lead to natural language feedback about the robot's performance that the robot should learn from. Finally, we discuss how the ability to summarize actions could help robots develop representations of environments and actions which could enable them to better adapt to new situations and tasks.

Blue Sky Papers, 5th Conference on Robot Learning (CoRL 2021), London, UK.

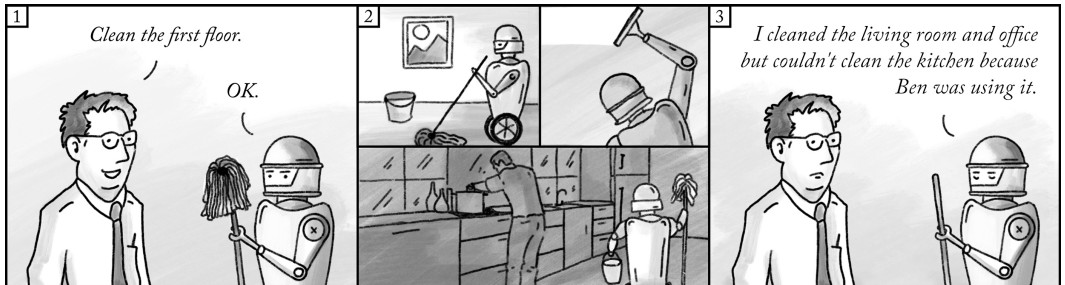

Figure 1: After being given instructions (panel one) a robot performs a series of actions (panel two) which it then summarizes (panel 3). The summary it gives illustrates pure summarization (task one) as well as a simple form of explanation (task two)

## 2 Robot action summarization: proposed tasks

### 2.1 Pure summarization of past actions

If a user sets a robot to perform a series of tasks, she would naturally want an easy to understand report afterwards. In some circumstances a detailed, technical readout of all actions taken might be appropriate but many situations call for natural language summaries, possibly spoken, e.g. in the home or in a busy work environment. Such summaries would ideally be:

1. Succinct. Reports should be summaries of actions taken, not a step-by-step recounting of every action taken. Even if a robot keeps a log of actions taken, learning how to generate a meaningful summarization at the right level of abstraction will be a challenge.

2. Thorough. Summaries should include all significant and relevant events that happened. Learning how to balance the inherent tension between this and the need to be succinct will also be difficult.

3. Context aware. The content of provided summaries should differ based on the context and should provide only information relevant to the current circumstances, to the extent the robot agent can have knowledge of those.

4. Inclusive of negative results. If a robot is unable to perform one or more of the tasks it is given, it should be able to include that failure in its summary.

### 2.2 Explanations of past actions

While a summary of actions taken would by itself be useful, the ability to also provide an explanation of events that occurred and actions that a robot took would be additionally beneficial. An explanation would be most helpful in two circumstances:

1. Failure. If a robot is unable to perform a task it would be much more informative to explain why it could not do so than simply report the bare fact of its failure. For example, if a cleaning robot was not able to wash the dishes, it could explain that its failure was due to the absence of sufficient dishwashing liquid.

2. If choices were made. If a robot is given a task to perform and such a task by its nature demands that the robot needs to make one or more choices during the execution of the task, it might (depending on circumstances) be important to know which choices were made and why. An explanation might be needed for why a certain route was taken to arrive at a destination, why a particular meal was prepared rather than another, or why a regimen of care was provided for a patient.

Figure 1 provides a simple example that shows a robot capable of performing tasks one and two (pure summarization and explanation). After being given instructions, it completes what it can of its jobs and then provides a quick summary of its actions, providing a reason why it could not accomplish everything it should have.

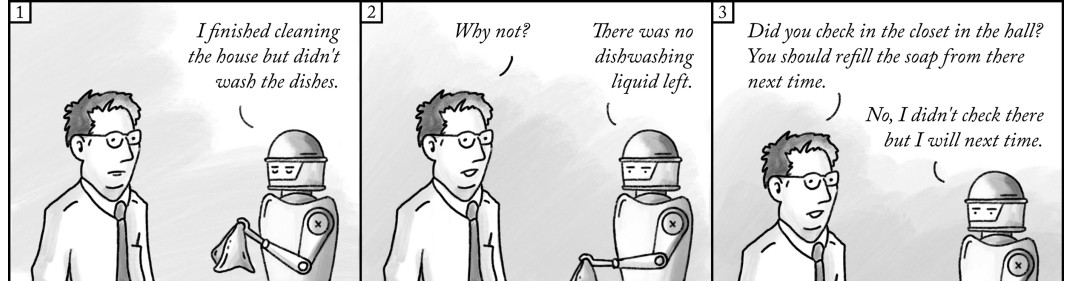

Figure 2: After reporting on its actions (panel one), a robot answers a question about its failure to complete a task (panel two) and then receives instructions for how to handle similar situations in the future (panel three). This series illustrates a robot engaged in question answering (task three) and learning from feedback to summarization (task four).

## 2.3   Question answering about past actions

After a robot provides a summary of its actions, it ought to be able to answer questions about those actions. Following our taxonomy of tasks, there are two primary areas of focus for question answering:

1. Factual. After hearing or reading a high-level summary, a user might have questions about the details of what a robot did. For example, after being told by a robot that it fed the cat, a user might ask what food the cat was fed.

2. Explanatory. A user's questions may move beyond the simple facts of what happened to the realm of explanation, asking a robot why it did something. A user might also come up with counterfactual questions, asking why a robot did not do something else. This kind of dialogue could naturally lead to feedback about what actions were taken, the reasons for those actions, or reasons to perform other actions in the future.

## 2.4   Learning from feedback to summaries

When trained in a reinforcement learning framework, it is standard for robots or agents in simulated environments to receive reward feedback based on whether, and possibly how, they actually completed a task assigned to them. While this is sensible, it might not always be practical. After sending a robot off to perform some actions, it is unrealistic to assume the user will continuously monitor the robot while it acts. Since many situations in which a robot could be useful involve the robot's acting outside a user's line of sight and/or for an extended period of time, direct supervision is challenging. However, if a robot were able to provide a summary of the actions it took, a user could provide feedback to that summary rather than the entire history of actions the robot actually performed. For example, a robot might report that it mopped and then swept the floor and receive natural language feedback from the user that it should instead first sweep and then mop floors.

Learning from feedback to summaries would be particularly useful in instances where the robot reported that it was unable to do something. In many cases, that inability might be due to the robot's encountering a situation in which it was unsure how to proceed. If it were able to provide an explanation of such a situation, a user could give it explicit instruction on how to act the next time.

Figure 2 illustrates a robot able to perform tasks three and four (question answering and learning from feedback to summaries). The robot answers a question by providing an explanation for why it could not complete a task and then receives corrective feedback telling it to do something different the next time a similar circumstance arises.

## 2.5   Developing the ability to summarize actions as a form of pretraining

The ability to report on and summarize its actions might improve a robot's performance on a variety of tasks even without considering the ability to learn from feedback to those reports. Learning what is worth reporting on would enable a robot to better understand its operating environments and users' expectations. It would constitute a form of pretraining to enable agents to better deal with new:

1. Environments. The ability to describe its actions in an environment could be used to adapt to new environments. Coming into a new environment with knowledge of what features of environments humans find important and what objects and affordances of environments might be worth including in a summary or answering questions about would serve as a kind of inductive bias to guide learning about operating in the new environment. It could help robots learn what sorts of things to "pay attention to" in a new environment.

2. Actions. Actions have a structure which can be explicitly or implicitly learned while learning how to summarize and answer questions about those actions. Summarizing actions can be used to develop an understanding of their compositional structure as an agent learns how low-level actions and their descriptions combine to form more meaningful high-level action descriptions.

## 3   Related Work

Our proposed research direction falls broadly within the area of human-robot interaction and more specifically at the burgeoning intersection of robotics and natural language processing [4, 5, 6, 7]. Some prior work has been done on providing detailed descriptions of robotic activities such as playing soccer [8] or automated driving [9, 10]. Language has also been used to clarify how actions were completed (e.g. naming which item was retrieved when told to pick something up) [11]. Our proposal is closely related to learning to follow natural language instructions, which has generated a great deal of interest, often in navigation tasks [12, 13, 14, 15, 16]. Several rich simulated environments for such tasks exist [17, 18]. Recent work has demonstrated that the ability to generate natural language instructions can improve an agent's performance on a variety of tasks [19, 20, 21].

Summarization of text has been a long-standing area of research within the natural language processing community [22, 23]. There are two primary approaches: extractive summarization, where some of the actual text that appears in a document is selected for inclusion in a summary; and abstractive summarization, in which new text is generated to serve as a summary. These suggest two possible avenues for summarizing a robot's action. First, a robot could generate and save a constant narrative of its actions, from which it later extracts the most important parts. Second, either with or without the contemporaneous narrative stream, it could generate more high-level descriptions which conceptually abstract away from the low-level details. The second, abstractive, approach would be preferable and must be the long-term goal but in the near term may be more challenging than the first. Summarization is far from solved even for pure text and the best summarization techniques have systematic flaws (e.g. they tend to favor including text based on its position in a document rather than its actual importance) [24], suggesting that a robot receiving real world multimodal input would face an even more challenging summarization task.

The tasks we propose touch on many other areas of robotics and machine learning, including human-robot dialogue [25, 26, 27], receiving feedback from natural language [28, 29], scene understanding [30, 31], action recognition [32], explainable robotics [33, 34], enabling agents to store and access memories [35, 36, 37], and using language in reinforcement learning [38, 39], including as way to improve transfer learning capabilities [40].

Recent proposals on the need to situate natural language processing in a grounded or embodied context [41] [42] [43] did not discuss robots' generating natural language descriptions or summaries of past actions, nor did a suggested roadmap for robots' use of spoken language [44]. Our proposal, then, extends and complements these.

## 4   Conclusion

Robots with the ability to summarize what they have done would be easier to supervise and cooperate with. They would better understand the environments they operate in and more easily adapt to new environments and tasks. Generating natural language summaries would open up a new channel for providing feedback to learn from. Summarization will not be an easy skill to acquire, requiring advances in many areas of robotics and machine learning as well as their integration. But the progress already made in those areas allows for directly tackling these challenges in the form of the tasks we propose.

**Acknowledgments**

We are grateful to Shuran Song and Iretiayo Akinola for helpful discussions. Illustrations were generously provided by Kai West Schlosser.

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
