# OpenReview forum: "Toward robots that learn to summarize their actions in natural language: a set of tasks"
_robot-learning.org/CoRL/2021/Conference/Blue_Sky — CoRL 2021, Blue Sky_

### Official Review · Reviewer_uzk5 · 2021-08-12

**Novelty:** Good
**Impact:** 2
**Clarity Of Presentation:** Good

**Recommendation:**

Weak Accept: I recommend accepting the paper, but will not argue for my recommendation if the majority of other reviewers have a different opinion.

**Summary:**

In this paper, the authors make the case for robot action summarization as the first component to focus on in the context of explainable robot learning. The authors also propose a set of challenges that will help establish the benchmarks for this problem.

**Summary Of Recommendation:**

The paper presents an interesting case for using action summarization as the first challenge to tackle in the context of explainable robots. Please see my points below:

- Section 2.4 raises great points that apply the described concepts to real world robotics such as the applicability of action summarization to reward functions. This is a novel and important point that has practical benefits
- The paper presents really good illustrative examples and figures
- It's a little unclear in Section 2.5 as to what such datasets would entail. Are these datasets with interactions and coupled explanations? Additional explanation would clarify this argument
- The manuscript doesn't outline any potential solutions to the problems and focuses on the challenge and potential benchmarks. It would be beneficial if the authors could provide some initial guidelines towards potential solutions.

The manuscript presents multiple valid points that justify action summarization as a first step towards the timely and important problem of explainable robot learning algorithms.

---

### Official Review · Reviewer_tzFT · 2021-08-29

**Novelty:** Good
**Impact:** 3
**Clarity Of Presentation:** Excellent

**Recommendation:**

Weak Accept: I recommend accepting the paper, but will not argue for my recommendation if the majority of other reviewers have a different opinion.

**Summary:**

The authors propose a new challenge for robotics research focused on enabling robots to summarize their past activities using natural language.  Addressing this research problem would bring together research from text summarization, human-robot interaction, natural language processing and explainable AI.  The paper proposes a series of increasingly complex subproblems for this research area, beginning from simple summarization, and progressing to explanations, interactive question answering, and learning from feedback.

The proposed problem is novel, I am not aware of any existing papers that address it from the perspective of past actions.  There is similar work out of the explainable AI planning community, namely by Chakraborti et al, which focuses on explaining sequential decision making tasks.  Their work, however, lacks the interpretable NLP aspect and only has been applied to the task the robot/agent is currently executing, so it is not the same as what is being proposed.

Overall, I agree with the authors that robot activity summarization is an important research problem.

It is also a somewhat natural progression of the current work in the research areas mentioned above.  I have heard several people talk about this area of research over the past year, so I don't know if it really qualifies as "Blue Sky", though given the lack of already published articles it seems to qualify.

**Summary Of Recommendation:**

The authors propose a new challenge for robotics research focused on enabling robots to summarize their past activities using natural language.

This is an interesting and important problem at the intersection of HRI, XAI, and NLP research.

A blue sky article may be useful to help bring together researchers across these communities.

---

### Decision · Program_Chairs · 2021-10-01

**Decision:**

Accept

**Comment:**

The paper presents a novel research agenda for explainable robot learning, esp., action summarization as a form of pre-training. Both reviewers are supportive, and I concur.